# Evaluation of Epigenetic Age Acceleration Scores and Their Associations with CVD-Related Phenotypes in a Population Cohort

**DOI:** 10.3390/biology12010068

**Published:** 2022-12-30

**Authors:** Olga Chervova, Elizabeth Chernysheva , Kseniia Panteleeva , Tyas Arum Widayati , Natalie Hrbkova, Jadesada Schneider , Vladimir Maximov , Andrew Ryabikov , Taavi Tillmann , Hynek Pikhart, Martin Bobak , Vitaly Voloshin , Sofia Malyutina , Stephan Beck 

**Affiliations:** 1UCL Cancer Institute, University College London, London WC1E 6DD, UK; 2Department of Pathology and Biomedical Science, University of Otago, P.O. Box 4345, Christchurch 8140, New Zealand; 3Institute of Internal and Preventive Medicine—Branch of Institute of Cytology and Genetics, Siberian Branch of Russian Academy of Sciences, Novosibirsk 630089, Russia; 4Institute of Family Medicine and Public Health, 50411 Tartu, Estonia; 5Institute of Epidemiology and Health Care, University College London, London WC1E 7HB, UK; 6Royal Botanical Gardens Kew, London TW9 3AE, UK

**Keywords:** DNAm age, epigenetic clock, epigenetic age acceleration

## Abstract

**Simple Summary:**

We consider a subset (*n* = 306) of an Eastern European ageing population cohort which was followed up for 15 years. Using blood DNA methylation data, we calculated nine epigenetic age acceleration scores, which are defined as deviations of epigenetic age from chronological age. We then evaluated how those scores are associated with available phenotypic data. This was implemented by splitting the phenotypic data into groups with positive and negative epigenetic age acceleration, and evaluating the difference between those groups. We observed strong association between all the considered epigenetic age acceleration and sex, suggesting that any analysis of these scores should be adjusted for sex. Moreover, even after adjusting for sex, the associations between the scores and considered phenotypes remain sex-specific. The only two associations that persisted through the entire dataset and both male and female subsets are incident coronary heart disease and smoking status. The observed associations of the various epigenetic age acceleration scores with both individual and groups of phenotypes suggest that these scores are sensitive to various cardiometabolic parameters, which might indicate their prognostic potential for related disorders.

**Abstract:**

We evaluated associations between nine epigenetic age acceleration (EAA) scores and 18 cardiometabolic phenotypes using an Eastern European ageing population cohort richly annotated for a diverse set of phenotypes (subsample, *n* = 306; aged 45–69 years). This was implemented by splitting the data into groups with positive and negative EAAs. We observed strong association between all EAA scores and sex, suggesting that any analysis of EAAs should be adjusted by sex. We found that some sex-adjusted EAA scores were significantly associated with several phenotypes such as blood levels of gamma-glutamyl transferase and low-density lipoprotein, smoking status, annual alcohol consumption, multiple carotid plaques, and incident coronary heart disease status (not necessarily the same phenotypes for different EAAs). We demonstrated that even after adjusting EAAs for sex, EAA–phenotype associations remain sex-specific, which should be taken into account in any downstream analysis involving EAAs. The obtained results suggest that in some EAA–phenotype associations, negative EAA scores (i.e., epigenetic age below chronological age) indicated more harmful phenotype values, which is counterintuitive. Among all considered epigenetic clocks, GrimAge was significantly associated with more phenotypes than any other EA scores in this Russian sample.

## 1. Introduction

It has been more than a decade since the very first epigenetic age predictor was proposed [1], and since then, dozens of DNA methylation (DNAm) based clocks have been developed. “Epigenetic age” (EA) is a score that is calculated by applying an EA prediction model (a DNAm clock) onto a set of DNA methylation measurements at particular loci (CpGs). Epigenetic age acceleration (EAA) is defined as the deviation of the estimated EA from the chronological age (CA), and is typically derived as either the difference between EA and CA or as the residual from regressing EA onto CA (EA∼CA).

In the beginning of the epigenetic clock era, the first-generation EA predictors (i.e., [1,2,3]) were primarily focused on accurate age prediction. The new EAA measures, which are derived from second-generation epigenetic clocks (i.e., [4,5]), are more focused on capturing physiological dysregulation [6] while still keeping strong links to chronological age [7].

Various measures of EAA are shown to be associated with different phenotypes and diseases (see reviews [8,9]). For example, deviations in EAA were shown to be connected to cancer [10,11], metabolic syndrome [12], and cognitive function decline [13]. All of these conditions are linked with ageing, which is a complex process that involves changes in all organs, tissues, and cells and cannot be quantified by a single biological measure. Similarly, there is no single EAA measure that could be declared as the best epigenetic marker of ageing.

In this study, we investigate the relationship between several widely-used EAA scores with the phenotypic data on cardiovascular disease (CVD) related risk factors and conditions available for a random population sample (n=306) that is a part of the Health, Alcohol, and Psychosocial Factors in Eastern Europe (HAPIEE) Project [14] is a Siberian cohort established in 2003 as a multicentre epidemiological study of CVD in Eastern and Central Europe. One of our aims was to determine which EAA measures are “sensitive” to which phenotypes and health-related conditions. By comparing the distributions of phenotypes between those with positive and negative EAAs, we identified how EAAs are associated with clinical data in an ageing Russian population.

## 2. Methods

### 2.1. Data Collection

This study is based on the data generated from a subset of the Russian branch of the HAPIEE (Health, Alcohol, and Psychosocial Factors in Eastern Europe) cohort [14], which was established in Novosibirsk (Russia) in 2003–2005 and followed up in 2006–2008 and then again in 2015–2017. The protocol of the baseline cohort examination included an assessment of cardiovascular and other chronic disease history, lifestyle habits and general health, socioeconomic circumstances, an objective measurement of blood pressure (BP), anthropometric parameters, physical performance, and instrumental measurement. The details of the protocol are reported elsewhere [14].

This study is based on a cohort of n=306 HAPIEE participants who did not have any indications of cardiovascular disease during the baseline measurement, as well as having a whole blood DNA methylation profile. DNAm was measured in accordance with the manufacturer’s recommended procedures using the Illumina MethylationEPIC BeadChip (Illumina, San Diego, CA, USA); a detailed description is available in [15].

### 2.2. Variables Description

All variables involved in our analyses were collected during the baseline examination (with the exception of incident coronary heart disease). Phenotypic data available for our study included age, sex, systolic and diastolic blood pressure values (SBP and DBP, mmHg; respectively), anthropometric parameters—body mass index (BMI, $\mathrm{kg}/m^{2}$) and waist–hip ratio (WHR, units), smoking status (ever smoker or never smoker), and the estimated annual alcohol intake (g of ethanol and number of annual occasions). A person who smoked at least one cigarette a day was classified as an “ever smoker”. The amount of alcohol consumed was assessed using the Graduated Frequency Questionnaire and was then converted to pure ethanol (g) [16]. The height and weight were measured with accuracy to 1 mm and 100 g, respectively. Blood pressure (BP) was measured three times (Omron M-5 tonometer) on the right arm in a sitting position after a 5 min rest period with 2 min interval between measurements. The average of three BP measurements was calculated and recorded.

Fasting blood serum test results contain measured levels of total cholesterol (TC, mmol/L), triglycerides (TG, mmol/L), high-density lipoprotein cholesterol (HDL, mmol/L), gamma-glutamyl transferase (GGT, mmol/L), and plasma glucose (mmol/L). The levels of TC, TG, HDL, GGT, and glucose in blood serum were measured enzymatically with the KoneLab 300i autoanalyser (Thermo Fisher Scientific Inc., Waltham, MA, USA) using Thermo Fisher Scientific kits. The Friedewald formula [17] was applied to calculate low-density lipoprotein cholesterol (LDL, mmol/L). Fasting plasma glucose (FPG) was calculated from the fasting serum glucose levels using the European Association for the Study of Diabetes (EASD) formula [18]. Hypertension (HT) comprises SBP≥140 mmHg or DBP≥90 mmHg according to the European Society of Cardiology/European Society of Hypertension (ESC/ESH) Guidelines [19] and/or antihypertensive medication intake within two weeks prior to the blood draw. Presence of Type 2 diabetes mellitus (T2DM) was defined as FPG>7.0 mmol/L, or ongoing treatment with insulin or oral hypoglycaemic medicines [20]. None of the participants included in our analysis had a history of major cardiovascular disease (CVD), such as myocardial infarction (MI), acute coronary syndrome (ACS), stroke, or transit ischemic attack at the time of the baseline examination and blood draw. The binary coronary heart disease (CHD) variable in our dataset includes any incident CHD events (MI/ACS) which occurred within the 15-year follow-up period of the cohort.

Carotid arteries were examined via high-resolution ultrasound using the systems Vivid q or Vivid7 (GE HealthCare) with a 7.5/10 mHz phased-array linear transducer. Device settings were adjusted in accordance with the American Society of Echocardiography (ASE) recommendations [21]. Longitudinal and transverse scans were performed at the right and left common carotid arteries with branches to assess anatomy and atherosclerotic lesions. The digital images were archived and the measurements were conducted offline by an experienced researcher (A.R.) who was blinded to the participants’ characteristics [22]. The plaques were defined in accordance with the Mannheim consensus [23]. For the present analysis, we used two phenotypes of atherosclerosis: presence of at least one carotid plaque (CP) or multiple plaques (MCP). The ultrasound variables are only available for a subset of samples (n=105, 35% of all samples).

Individual phenotypes were also combined into five groups of phenotypes, which we define as follows:
**Anthropometric:** BMI and WHR;**Lifestyle:** smoking status and annual alcohol consumption (intake and number of occasions);**Metabolic:** GGT, T2DM, and plasma glucose;**Lipids:** TC, HDL, LDL, and TG;**Cardiovascular:** SBP, DBP, HT, CHD, CP, and MCP.

### 2.3. DNAm Data Quality Control (QC) and Preprocessing

In preprocessing raw DNAm data, we mostly followed the procedures from [24], which are in line with the manufacturer’s recommended steps. In brief, we checked the array control probes’ metrics (Illumina Bead Control Reporter), signal detection *p*-values, and bead count numbers for all available cytosine–phosphate–guanine (CpG) probes. Furthermore, we compared actual and DNAm predicted sex data for each sample. Samples included in this analysis have less than 1% of CpGs with detection p≥0.01 and bead count number ≥3, all the included probes have detection p<0.01 in at least 99% of samples. Initial DNAm data processing and QC data filtering were implemented using R v.4.1.0 [25] together with specialised R libraries minfi [26], ChAMP [27], and ENmix [28].

### 2.4. Epigenetic Age Acceleration

EAA scores were calculated using the DNA Methylation Online Calculator [3]. This web-based tool gives nine EAAs based on five epigenetic scores, namely, Horvath’s [3], Hannum’s [2], Skin and Blood [29], PhenoAge [4], and GrimAge [5] measures; see Table 1. Further details regarding epigenetic clocks and various EAAs are given in Appendix B.

### 2.5. Grouping

In this study, we evaluated CVD-related phenotypes and their association with different EAA scores. It is expected that these phenotypes show small effect size (in comparison with some types of cancer) in blood DNA methylation, and hence in EAAs as well. Taking into account the relatively small sample size of our study, we decided to limit our analyses to the grouping of EAAs as described below.

Analysis of associations between EAAs and phenotypes in our study involves comparing the distributions of the phenotypic data in two groups. The grouping is based on binary split with respect to the sign of EAA, defined as follows:(1)$$\mathrm{All}\;\mathrm{samples}=\left\{\begin{array}{cc} {\mathrm{EAA}}^{+}, & \mathrm{samples}\;\mathrm{with}\;\mathrm{non}-\mathrm{negative}\;\mathrm{EAA},\\ {\mathrm{EAA}}^{-}, & \mathrm{samples}\;\mathrm{with}\;\mathrm{negative}\;\mathrm{EAA}. \end{array}\right.$$

In other words, for each clock we use the definition (Equation 1) to split our cohort into two groups, one with EAA<0 and the other with EAA≥0, and then study the differences in phenotypic distribution between these groups. Similar grouping was also featured in previous studies based on EAAs [31,32].

### 2.6. Statistical Analysis

All statistical analyses were performed using R v.4.1.2. They include descriptive analysis of the available data using relevant techniques, such as univariate analysis, cross-tabulation, statistical hypothesis testing (Welch’s *t*-test [33] for continuous variables and Fisher’s exact test [34] for binary data), and linear-regression-based data adjustments. Welch’s *t*-test null hypothesis: mean values of a given variable in ${\mathrm{EAA}}^{+}$ and ${\mathrm{EAA}}^{-}$ groups are not different. Fisher’s exact test null hypothesis: classifications of a given binary variable in ${\mathrm{EAA}}^{+}$ and ${\mathrm{EAA}}^{-}$ groups are not different. The significance level is defined as α=0.05 for each EAA–phenotype association hypothesis test.

In order to consider the association between different EAAs and groups of phenotypes (all apart from the lifestyle group), we controlled for family-wise error rate (FWER) using the Bonferroni correction [35,36], which was performed per group of phenotypes per EAA. The significance threshold for the anthropometric, metabolic, lipids, and cardiovascular groups of phenotypes were calculated to be 0.025, 0.0166, 0.0125, and 0.0083, respectively. In the lifestyle group, we considered the smoking and alcohol intake data separately; thus, FWER-controlled significance threshold for alcohol consumption phenotypes is 0.025, and 0.05 for smoking status. It means that for each clock, the group association was inferred from the individual phenotypes by controlling for FWER in the different phenotype groups. In other words, we define the EAA score to be associated with the group of phenotypes if for at least one of the phenotypes in the group the significance of the relationship is sustained with the Bonferroni-corrected threshold.

All the graphs presented in the paper were produced using ggplot2 [37] and its extensions, pheatmap [38], PerformanceAnalytics [39], and base R functions.

## 3. Results

### 3.1. Associations between Sex and Phenotypes

Our dataset consisted of (n=306) samples (166 females and 140 males). Summaries of the dataset characteristics for all samples and for sex-specific groups are given in Table A1 and Table A3. Table A1 contains descriptive statistics (range, mean, and standard deviation) for the available continuous phenotype data and the corresponding Welch’s *t*-test *p*-values and 95% confidence intervals. Table A3 includes count numbers and percentages for dichotomous variables, together with sex-specific odds ratios, 95% confidence intervals, and *p*-values calculated by performing a Fisher’s exact statistical test.

The Russian sample being considered in this study showed no significant difference between males and females in the distribution of chronological age, blood pressure values (both SBP and DBP), incidence of acute CHD events, diagnosis of hypertension and diabetes, or levels of triglycerides and fasting glucose. The phenotypes which were significantly different in males vs. females are anthropometric measures (BMI and WHR), lifestyle choices (alcohol consumption and smoking status), and blood levels of gamma-glutamyl transferase (GGT) and lipids (both LDL and HDL). Interestingly, in this Russian dataset, there was no significant difference between the male and female odds ratios of being diagnosed with a carotid plaque (CP), but the odds ratios of having multiple carotid plaques (MCP) significantly differed between sexes.

### 3.2. EAAs Are Associated with Some Phenotypes and Have Strong Sex Bias

Our EAA analyses are based on nine EAA scores (described in Section 2.4) which were obtained from the five different epigenetic clock models, with multiple EAA scores derived from Horvath’s multi-tissue and Hannum’s clocks (three EAAs each). Correlation coefficients are higher among EAAs based on the same clock than among EAAs from different clocks. Namely, Pearson correlation coefficients range between 0.78 and 0.97 within EAAs derived from Horvath’s and Hannum’s models, whilst the highest value of correlation for EAAs derived from separate clocks is r=0.57 (see correlation table in Figure A2). Note that neither of the EAAs is significantly correlated with chronological age apart from HorvathAAd, which is the only measure calculated without chronological age adjustment.

To explore connections among the variables, we calculated correlation coefficients (Spearman correlation) and normalised entropy-based mutual information values for all the phenotypes and EAAs. Heatmaps for correlations (absolute values) and mutual information values, as well as correlations-based network plots, are presented in corresponding Figure 1, Figure A1 and Figure 2, respectively. In both the correlation and mutual information plots, the EAAs are clustered together with the exception of GrimAA, which displays very strong associations with sex and smoking status.

We further investigated the relationship between phenotypes and EAAs by splitting the dataset into ${\mathrm{EAA}}^{+}$ and ${\mathrm{EAA}}^{-}$ groups using (Equation 1), and, subsequently, testing the phenotype data distribution using a *t*-test for continuous variables and Fisher’s exact test for binary variables. The corresponding statistical testing results are presented in Table A5 and Table A6.

We noted that in nearly all EAA measures the size distribution of the ${\mathrm{EAA}}^{+}$ and ${\mathrm{EAA}}^{-}$ groups were within a 45–55% range. The only exceptions to this were HorvathAAd (32%
${\mathrm{EAA}}^{+}$ samples vs. 68%
${\mathrm{EAA}}^{-}$ samples) and GrimAA (61%
${\mathrm{EAA}}^{+}$ samples vs. 39%
${\mathrm{EAA}}^{-}$ samples). Sex-specific group splitting was found to be very unbalanced for all the EAAs for both sexes with the exception of HorvathIEAA (see Table A4). Furthermore, we observed significant differences in distributions of all nine EAA measures in our data between males and females; the corresponding data along with descriptive statistics are presented in Table A2. Given the strong association between sex and the various phenotypes examined in this study, the significant results obtained for EAA–phenotype associations might have been confounded by sex.

### 3.3. Sex-Adjusted EAAs Are Associated with Various Phenotypes

In order to eliminate the undesired sex bias, we adjusted all of the EAA scores by sex and then repeated the analyses described in the previous section, based on the calculated adjusted EAAs (adjEAA). Splitting the data into ${\mathrm{EAA}}^{+}$ and ${\mathrm{EAA}}^{-}$ resulted in balanced group sizes for all the adjEAAs, and all of the groupings were within a 44–56% range; see Table A4. The medians of the adjEAAs for sex-specific subsets located closer to 0 compared to the medians of unadjusted EAAs (see Figure 3A).

The significant results from testing the differences in phenotype distribution between the ${\mathrm{EAA}}^{+}$ and ${\mathrm{EAA}}^{-}$ groups are given in Table 2. This table contains 95% confidence intervals, which indicate the trends in the direction of differences. The corresponding means and odds ratio values could be found in Table A6, where we present all testing outcomes regardless of their significance. For all available samples, only four adjEAAs (GrimAA, PhenoAA, Horvath’s residuals, and IEAA) demonstrated statistically significant results for six phenotypes, with four phenotypes highlighted by GrimAA, and one phenotype each by the rest of the adjAAs (seven phenotype–EAA combinations in total). Among the differently distributed phenotypes are blood levels of GGT and LDL, smoking status and annual alcohol consumption, diagnosed MCP, and incident CHD status, with the latter being the only phenotype that tested significantly different by multiple adjEAAs (GrimAA and Horvath’s differences). Interestingly, for the GrimAA clock, incident CHD and smoking status stayed significantly different for both male-only and female-only subsets, whilst GGT was not significantly different in any sex-specific groupings.

Seven phenotype–EAA combinations were demonstrated to be statistically significant. Of these, five remained significant in male-only data subsets, and three remained significant in female-only subsets. In males, the significant differences between the ${\mathrm{EAA}}^{+}$ and ${\mathrm{EAA}}^{-}$ groups were confirmed by four EAAs (the same for all samples) and seven phenotypes (10 phenotype–EAA combinations). Significant results for females feature seven EAAs (all apart from Horvath’s and HannumIEAA) and 10 traits (21 phenotype–EAA combinations). Nearly half (10 out of 21) of the results for the female subgroup presented in Table 2 relate to blood lipids measures (TG, LDL, and HDL), and another six results for females relate to the presence of a hypertension diagnosis and blood pressure values (SBP and DBP). Neither lipid- nor blood-pressure-related phenotypes were associated with the ${\mathrm{EAA}}^{+}/{\mathrm{EAA}}^{-}$ grouping in males, unlike the presence of a CP/MCP diagnosis. Anthropometric parameters were also found to be statistically different in both sex-specific groups (BMI in males and WHR in females), but not for the combined dataset.

Our findings also suggest that some of the clocks are associated with groups of phenotypes. In particular, we observed that GrimAA and HorvathAAr are significantly associated with the cardiovascular group of phenotypes for the entire dataset. In the female-only subset, HorvathAAd and GrimAA are both associated with the anthropometric and cardiovascular groups, whilst PhenoAA, HannumAA, and HannumEEAA are associated with the lipids group. No significant results were found for the metabolic group or in the male-only subset.

### 3.4. Directions of Some EAA–Phenotype Associations in Sex-Specific Subsets Are Different

The results presented in Table 2 and Table 3 summarise the significant outcomes of statistical hypothesis testing of our data based on grouping (Equation 1). Some of the phenotypes in the sex-specific groupings were highlighted by multiple EAAs, e.g., female total cholesterol (HannumAA, PhenoAA, GrimAA, and EEAA) and male alcohol consumption (IEAA and GrimAA); however, the signs of the groups’ mean differences are not consistent.

For instance, in Table 2, the confidence intervals for female WHR are positive for GrimAA and negative for HorvathAAd. Further investigation revealed that the ${\mathrm{EAA}}^{+}$ group have a higher mean WHR than in ${\mathrm{EAA}}^{-}$ for GrimAA, but the opposite is true for HorvathAAd (see Figure A6). Furthermore, the mean WHR values were higher in the ${\mathrm{EAA}}^{-}$ group for all three EAAs derived from Horvath’s clock, together with HannumIEAA and SkinBloodAA. Similar trends were observed in male annual alcohol consumption (see Figure A3) and in female levels of TC, HDL, LDL, and blood pressure values (SBP and DBP) (see Figure A7, Figure A8, Figure A9, Figure A11 and Figure A12, respectively).

## 4. Discussion

The question of which EAA measure is the “best” or “most suitable” to study particular phenotypes is yet to be answered. For our data, we decided to take into consideration all the EAA measures that could be calculated using DNA Methylation Online Calculator [3], which is a relatively easy-to-use open-access tool. Epigenetic clocks included in the Online Calculator are featured in the vast majority of studies related to EAA–phenotype/disease associations (see, e.g., [32,40,41,42]), as well as in benchmarking the newly developed DNAm-based clocks’ performance (see, e.g., [43]).

Ability of the considered nine EAAs to reflect the differences in phenotype distribution was investigated by splitting the data based on the sign of the EAA scores. Similar grouping was also used in [31], where the risk of CHD was studied by splitting HorvathAAd and HannumAA into positive and negative groups. In a recent paper [32], the authors used the positive/negative GrimAA and PhenoAA split to study the incident diabetes in the Coronary Artery Risk Development in Young Adults (CARDIA) cohort.

All the considered EAAs were independent (apart from HorvathAAd) of chronological age, but clearly sex-biased (Table A2), with generally lower EAA values for females. It was particularly obvious for the GrimAA scores, with distribution profiles separated for males and females; see Figure 3B,C. As we pointed out in Section 3.3, splitting the dataset into ${\mathrm{EAA}}^{+}$ and ${\mathrm{EAA}}^{-}$ groups revealed big variation in group sizes for different scores (Table A4), which became particularly extreme for sex-specific subsets. To avoid unwanted confounding, for our analyses we adjusted EAAs by sex and proceeded with adjEAA values. This step resulted in a more balanced ${\mathrm{EAA}}^{+}/{\mathrm{EAA}}^{-}$ group split for all adjEAAs. Of course, adjusting EAAs for sex did not affect the actual differences in phenotypes distributions between male and female subjects (see Table A1 and Table A3). As a result, several phenotype–EAA combinations, which have previously demonstrated statistically significant results, did not persist after the adjustment (see Table A5).

Due to some phenotypes showing sex-specific behaviour (see, e.g., [44,45,46]), we presented the results for males and females separately alongside the results for the entire dataset. In one of the recent reviews [47], the authors pointed out the lack of sex-specific results involving EAAs and recommended splitting data by sex in downstream analyses. Our findings confirm the importance of using EAAs in sex-specific groups. We observed that most phenotypes are reflected by some EAAs in one sex-specific group only. It should be noted that the results presented used positive and negative EAAs grouping (see Section 2.5). The observed association in groups could be further investigated in larger datasets using continuous EAAs.

We found that in our dataset both BMI and WHR were significantly different in males and females. Without adjusting for sex, multiple EAAs groupings highlighted significant differences in both BMI and WHR for all samples, but none of those associations were replicated after adjustment (see Table A5). It is known [44] that in females WHR is associated with risks of CHD regardless of BMI, whilst in males, WHR was found to be associated with incidence of CHD only for subjects with normal BMI measures. Our analysis found the anthropometric parameters to be statistically different in both sex-specific groups (BMI (GrimAA) in males and WHR (GrimAA and HorvathAA) difference in females), which is in line with results reported in large-scale US Sisters study [48] and Taiwan Biobank [49] cohorts. Interestingly (and opposite to findings in [44]), in [49], the authors reported significant associations between WHR and EAAs (PhenoAge and GrimAge) in males, and between BMI and EAAs (PhenoAge and GrimAge) in females, which is the other way around in a studied Russian sample (WHR in females and BMI in males). We would like to point out that the GrimAA grouping revealed higher mean WHR in female ${\mathrm{EAA}}^{+}$, but lower male mean BMI in the same group with positive EAAs (see Figure A5 and Figure A6).

Lifestyle habits, including diet, smoking, and alcohol consumption, are known to impact DNAm and are associated with epigenetic age in multiple studies (see, e.g., [50,51,52]). Some DNAm clocks were specifically developed to be sensitive to smoking status, such as, for example, GrimAge [5]. GrimAA was the only score associated with smoking status in the entire dataset, and the associations were replicated in sex-specific subsets (Table 2). PhenoAA and HorvathAAd were also found to be significantly associated with smoking status in male and female subgroups, respectively. We would like to point out the very uneven distribution of the smoking status in men and women in our cohort (see Table A3), which is in line with the published data for the Russian population [53], but should be taken into account when comparing our findings with the results in other populations. HorvathIEAA was significantly associated with annual alcohol consumption for all the samples and this association persisted in males, together with GrimAA, but not in females. Interestingly, the mean annual alcohol volume was higher in the ${\mathrm{EAA}}^{+}$ group for GrimAA, but lower in the ${\mathrm{EAA}}^{+}$ group for IEAA (Figure A3), which is not in line with the current state of the art in alcohol–ageing relationships (see, e.g., review [54]).

Previous publications suggest that EAAs are associated with diabetes and/or glucose levels [9,55,56]. It was also found that positive GrimAA (but not PhenoAA) is associated with a 5–10 years’ higherincidence of Type 2 diabetes, particularly for obese individuals [32]. In analysing this Russian sample, we did not observe any significant associations of the considered EAAs with prevalent Type 2 diabetes mellitus (T2DM) status and/or fasting blood glucose values. This might be attributed to the small proportion (11%) of the diabetics in our data compared to other studies (e.g., nearly 20% in [56]). Blood levels of GGT are associated with many dysmetabolic conditions, including fatty liver, excessive alcohol consumption, and increased risks of CHD and T2DM [57,58], and is known to be different in men and women [45], with no unified reference values. For the entire dataset, GrimAA ${\mathrm{EAA}}^{+}/{\mathrm{EAA}}^{-}$ grouping demonstrated significant difference in serum GGT measures. This result was not replicated in sex-specific subsets, but at the same time, in the male subgroup, a GGT-leveldifference was detected in HorvathAAr split (see Figure A4).

Blood levels of total cholesterol, TG, and lipoproteins (HDL and LDL) are known to be sex-specific and associated with risk of developing CVD in both sexes (see, e.g., [59]). Changes in lipids concentrations are also shown to be reflected in age-related changes in DNAm following dietary interventions [60]. Furthermore, associations of EAAs and lipids levels were confirmed in several studies [61,62]. In our entire dataset, among all available lipids data, only mean LDL levels in ${\mathrm{EAA}}^{+}$ with PhenoAA grouping were significantly lower than in ${\mathrm{EAA}}^{-}$, and this result persisted in the female subset. No significant differences in mean lipids concentrations were highlighted by any EAA split for the male subgroup, whilst ten EAA–lipids phenotypes associations were highlighted in females. In particular, in the female subset, GrimAA grouping demonstrated significantly higher group mean levels of total cholesterol and TG in ${\mathrm{EAA}}^{+}$ compared to ${\mathrm{EAA}}^{-}$ (see Table 2, Figure A9 and Figure A10). At the same time, mean TC and LDL concentrations (Figure A7) were significantly lower in the ${\mathrm{EAA}}^{+}$ group in PhenoAA, HannumAA, and EEAA splits. Female HDL levels associations were picked up in SkinBloodAA and HorvathAAr groupings, with higher lipoprotein concentration in the ${\mathrm{EAA}}^{+}$ group (see Figure A8). Remarkably, for all four considered lipids-related measures, known CVD risk factors (high TC, LDL, and TG, and low HDL) were associated (not all significantly) with positive age acceleration only for the GrimAA grouping, whilst the opposite was demonstrated in all the significant (and vast majority of insignificant) EAAs–lipids associations based on other EAA splits (see Figure A7, Figure A8, Figure A9 and Figure A10). In view of recently published age-related sex-specific trends in lipid levels [63] and hypertension prevalence [46], it would be interesting to conduct extended sex-specific analyses on EAA–lipids and hypertension associations for the particular age groups to see whether EAA values reflect the observed age-related patterns.

Data on carotid atherosclerosis and advanced atherosclerosis, which are defined in our study as the presence of at least a single (CP) and multiple carotid plaques (MCP), respectively, were available for only 34% of the participants, with 50/23/14 and 55/23/2 total/CP/MCP samples available for males and females. Only the GrimAA grouping was significantly associated with CP in males and MCP in the entire dataset and its male-only subset. In female-specific subset, blood pressure values (both SBP and DBP) and hypertension status were significantly associated with HorvathAAd and GrimAA groupings. Interestingly, in the case of the GrimAA group split, mean values of SBP and DBP were higher in the ${\mathrm{EAA}}^{+}$ group, which might indicate the increased risk of CVD [64]. This is the opposite to the corresponding results of HorvathAAd grouping. None of these phenotypes were highlighted in the entire dataset and male subset. Two groupings, GrimAA and HorvathAAr, were significantly associated with incident CHD for all available samples. The results persisted in the male subset for both groupings and in the female subset for GrimAA split only. Similar results were also described in the Genetic Epidemiology Network of Arteriopathy (GENOA) dataset study [56], where the authors reported significant connections not only between GrimAA and incident CVD, but also between GrimAA and time to the CVD event.

Notably, while higher odds of CHD were associated with ${\mathrm{EAA}}^{+}$ for both GrimAA and HorvathAAr, only GrimAA ${\mathrm{EAA}}^{+}$ was consistently associated with more harmful phenotypes values, indicating higher risk of CHD. All other EAA splits demonstrated mostly the opposite behaviour regarding available risk factors (lipids, anthropometric, lifestyle, and cardiovascular).

The observed associations of the various EAAs with both individual and groups of phenotypes suggest that EAA scores are sensitive to various cardiometabolic parameters, which might indicate their prognostic potential for related disorders. Further investigations conducted on well-annotated larger datasets are needed to improve the understanding of the mechanisms behind those associations and to possibly develop new biomarkers. These might be extended by applying other epigenetic age models and using continuous EAAs in association studies.

## 5. Conclusions

Our study, conducted on a subset of HAPIEE cohort, shows that EAAs are sex-specific and should be adjusted for sex in EAA–phenotypes association studies. Moreover, even after adjusting for sex, the associations between EAAs and 18 considered cardiometabolic phenotypes are sex-specific. The only two phenotype–EAA associations that persisted through the entire dataset and both male and female subsets are incident CHD and smoking status.

Among all considered epigenetic clocks, GrimAge was significantly associated with more phenotypes than any other EA scores, but for most of the phenotypes, those associations are weaker than in other scores. Furthermore, for some EAAs, the direction of the association with phenotype is counterintuitive, i.e., lower EAA scores corresponded to more harmful values of the phenotypes. The observed associations of the various EAAs with both individual and groups of phenotypes suggest that EAA scores are sensitive to various cardiometabolic parameters, which might indicate their prognostic potential for related disorders.

## Figures and Tables

**Figure 1 biology-12-00068-f001:**
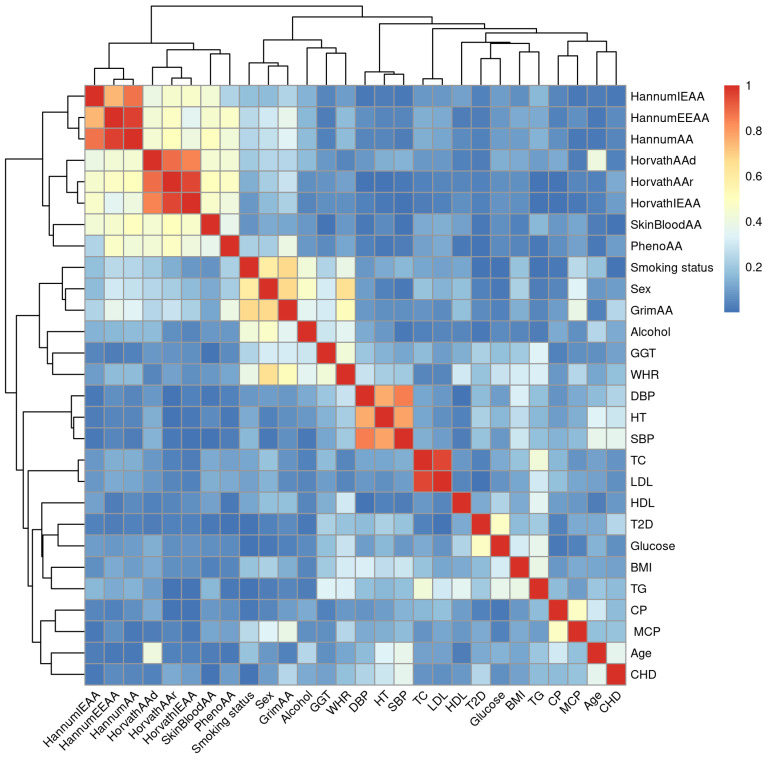
Heatmap of the correlations between all available traits and epigenetic age accelerations, based on Spearman correlation and unsupervised clustering.

**Figure 2 biology-12-00068-f002:**
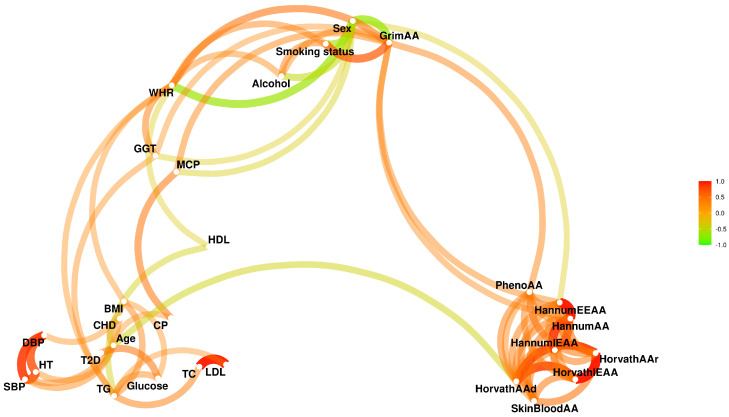
Network plot of of the connections among the phenotypes and EAAs in the dataset, based on Spearman correlation coefficients with absolute values above 0.3.

**Figure 3 biology-12-00068-f003:**
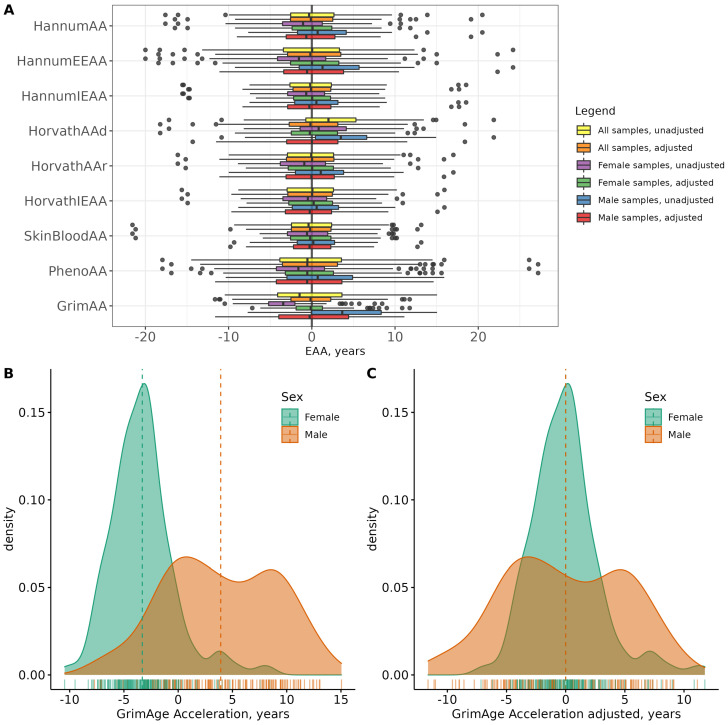
Boxplots of EAAs and sex-adjusted EAAs for all samples and sex-specific subsets (panel (**A**)), distribution of GrimAA distribution stratified by sex (panel (**B**)) and adjusted for sex (panel (**C**)). Dashed vertical lines on panels (**B**,**C**) correspond to group means.

**Table 1 biology-12-00068-t001:** **Summary of the EAA scores measured using DNA Methylation Online Calculator.*****Abbreviations: CA**—chronological age, **EA**—epigenetic age, **EAA**—epigenetic age acceleration, **IEAA**—intrinsic epigenetic age acceleration, **EEAA**—extrinsic epigenetic age acceleration*.

EAA	Clock	Info
**HannumAA**	Hannum [2]	Residuals from regressing EA on CA
**HannumEEAA**	Hannum [30]	Residuals from regressing the weighted average of Hannum’s EA and estimated measures of blood cells counts on CA
**HannumIEAA**	Hannum [30]	Residuals from regressing Hannum’s EA on CA and various blood immune cell counts
**HorvathAAd**	Horvath [3]	Difference between EA and CA
**HorvathAAr**	Horvath [3]	Residuals from regressing EA on CA
**HorvathIEAA**	Horvath [30]	Residuals from regressing Horvath’s EA on CA and various blood immune cell counts
**SkinBloodAA**	Skin and Blood [29]	Residuals from regressing EA on CA
**PhenoAA**	PhenoAge [4]	Residuals from regressing EA on CA
**GrimAA**	GrimAge [5]	Residuals from regressing EA on CA

**Table 2 biology-12-00068-t002:** **Significant differences between groups with positive and negative epigenetic age acceleration scores (${\mathrm{EAA}}^{+}$ and ${\mathrm{EAA}}^{-}$).** Results which remained significant after controlling for family-wise error rate of 0.05 in groups of phenotypes per clock (as described in the Methods Section 2.6) are presented in **bold**.

Phenotype	EAA	*p*, All	95% CI, All	*p*, F	95% CI, F	*p*, M	95% CI, M
* **Anthropometric** *							
**BMI**	GrimAA					0.039	(−3.079, −0.079)
**WHR**	HorvathAAd			**0.004**	(−0.048, −0.009)		
	GrimAA			**0.010**	(0.006, 0.046)		
* **Lifestyle** *							
**Smoking status**	GrimAA	**<0.001**	(1.799, 4.895)	**0.026**	(1.077, 8.931)	**<0.001**	(4.5, 58.7)
	HorvathAAd			**0.016**	(1.140, 9.454)		
	PhenoAA					**0.004**	(1.360, 8.319)
**Alcohol,** annual intake	HorvathIEAA	0.028	(−2832, −163 )			**0.023**	(−5522, −422)
	GrimAA					0.049	(16, 5370)
* **Metabolic** *							
**GGT**	GrimAA	0.023	(0.728, 9.699)				
	HorvathAAr					0.030	(0.738, 14.5)
* **Lipids** *							
**TC**	HannumAA			**0.009**	(−0.947, −0.141)		
	GrimAA			0.046	(0.008, 0.818)		
	PhenoAA			**0.010**	(−0.919, −0.127)		
	HannumEEAA			**0.003**	(−1.004, −0.203)		
**TG**	GrimAA			0.015	(0.070, 0.632)		
**HDL**	HorvathAAr			0.013	(0.026, 0.219)		
	SkinBloodAA			0.027	(0.012, 0.205)		
**LDL**	PhenoAA	0.037	(−0.523, −0.016)	**0.004**	(−0.840, −0.157)		
	HannumAA			**0.010**	(−0.811, −0.112)		
	HannumEEAA			**0.002**	(−0.904, −0.215)		
* **Cardiovascular** *							
**CHD**	GrimAA	**<0.001**	(1.518, 4.060)	**0.001**	(1.458, 5.955)	0.042	(1.020, 4.389)
	HorvathAAr	**0.006**	(1.187, 3.139)			0.018	(1.150, 4.995)
**CP**	GrimAA					0.009	(1.367, 22.723)
**MCP**	GrimAA	**0.004**	(1.584, 26.779)			0.009	(1.401, 33.864)
**HT**	HorvathAAd			**0.005**	(0.202, 0.781)		
	GrimAA			0.043	(0.987, 3.764)		
**SBP**	HorvathAAd			**0.008**	(−20.1, −3.2)		
	GrimAA			0.024	(1.3, 18.4)		
**DBP**	HorvathAAd			**0.003**	(−10.626, −2.144)		
	GrimAA			0.039	(0.228, 8.832)		

**Table 3 biology-12-00068-t003:** **EAA–phenotype association table.** White colour indicates no significant association, green/red colour indicate significantly higher/lower values of phenotype measures (higher odds ratios) in ${\mathrm{EAA}}^{+}$ group compared to ${\mathrm{EAA}}^{-}$ for continuous (binary) phenotypes. Lower case letters indicate individual significant associations between a clock and a phenotype for all cohort participants (a), females (f), and males (m); capital letters indicate phenotype group significant association (controlled for family-wise error rate) for all (**A**), females (**F**), and males (**M**).

	HannumIEAA	HannumEEAA	HannumAA	HorvathAAd	HorvathAAr	HorvathIEAA	SkinBloodAA	PhenoAA	GrimAA
**BMI**									m
**WHR**				**F**					**F**
**Smoking Status**				**F**				**M**	**AFM**
**Alcohol** (annual intake)						a **M**			m
**Alcohol** (annual occasions)									
**GGT**					m				a
**TC**		**F**	**F**					**F**	f
**TG**									f
**HDL**					f		f		
**LDL**		**F**	**F**					a**F**	
**CHD**					**A** m				**AF**m
**CP**									m
**MCP**									**A** m
**HT**				**F**					f
**SBP**				**F**					f
**DBP**				**F**					f

## Data Availability

Raw DNA methylation IDAT files have been deposited at the European Genome-phenome Archive (EGA), which is hosted by the EBI and the CRG, under accession number EGAS00001006390. The dataset will become accessible in July 2023. Further information about the EGA can be found at https://ega-archive.org (accessed on 19 December 2022): “The European Genome-phenome Archive of human data consented for biomedical research” [65].

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
