# Peer review of "Evaluation of Epigenetic Age Acceleration Scores and Their Associations with CVD-Related Phenotypes in a Population Cohort"

_biology, 2022, doi:10.3390/biology12010068_

Round 1

Reviewer 1 Report

This is a well-written paper, presenting a comprehensive analysis in a small cohort comparing phenotypes with various measures of epigenetic clocks. The data is clearly presented. The conclusions are measured. Several avenues for future research are indicated as well as some of the possible sources of errors. Appendix B was appreciated.

My only concern is that there are quite substantial differences between the male and female cohorts based on smoking status. In Table A3, it is noted that 14.46% of females smoked compared to 72.86 of the males. Interestingly, this phenotype is not addressed comparably to alcohol intake on Table A1. The authors address smoking status frequently in the manuscript and link it to EAA+. I regard the study to be valuable, but the single differential of smoking status is a highly significant one concerning the applicability of this study to a broader population that is not similarly skewed as this one well-documented major health risk. This factor should be acknowledged as a potential source of error by the authors, and it would be reasonable for them to speculate how this might have impacted their data.

Minor notes:

Line 294, change 'show' to 'showing'

Line 337, change 'dismetabolic' to 'dysmetabolic'

Author Response

We thank the reviewer for their valuable comments and suggestions.

Following the reviewer's advice, we made the following edits to the manuscript.

  1. We corrected the spelling mistakes spotted by the reviewer (and a few additional ones).
  2. We appreciate the reviewer's concern about the uneven distribution of smoking status in males and females. To address this concern we added the following lines to the Discussion section: "We would like to point out the very uneven distribution of the smoking status in men and women in our cohort (see Table A3), which is in line with the published data for the Russian population [53], but should be taken into account while comparing our findings with the results in other populations."

Reviewer 2 Report

In this study, Chervova et al. analyzed the association between epigenetic age acceleration scores and cardio-metabolic phenotypes by using the data generated from a subset of the Russian branch of the HAPIEE (Health, Alcohol, and Psychosocial Factors in Eastern Europe). They found that strong association between all epigenetic age acceleration scores and sex. In addition, some sex-adjusted epigenetic age acceleration scores were significantly associated with several phenotypes such as blood levels of gamma-glutamyl transferase and low-density lipoprotein, smoking status, annual alcohol consumption, multiple carotid plaques, and incident coronary heart disease status. This study is well-designed, and the results are quite convincing and informative. The manuscript is well-written. Some suggestions are shown below.

1. Full name of some abbreviations should be shown when first mentioned. For example, the full name of HAPIEE (line 48) should be mentioned where HAPIEE was first used (line 39). Abbreviations, but not the full names, should be used afterward in the manuscript. For example, EAA is the abbreviation for epigenetic age acceleration (line 1 in abstract). However, both EAA and “epigenetic age acceleration” were appeared many times in the manuscript. Authors should go through their manuscript and correct these non-standard issues.

2. It will be better to improve some of the figure legends with detailed information for better understanding.

Author Response

We thank the reviewer for their feedback and suggestions, which definitely helped to improve the manuscript.

Following the reviewer's comments, we made the following edits.

  1. We expanded the abbreviation for HAPIEE in line 39, and removed the expansion from line 48. In addition, following the reviewer's advice, we checked the use of EAA and epigenetic age acceleration throughout the manuscript and corrected it accordingly.
  2. We thank the reviewer for the suggestion, which clearly reflects the reader's experience. We expanded the captions of several figures (Figures 1-3) in the manuscript to ensure their clarity.